# Advancing the Robotic Vision Revolution: Development and Evaluation of a Bionic Binocular System for Enhanced Robotic Vision

**DOI:** 10.3390/biomimetics9060371

**Published:** 2024-06-19

**Authors:** Hongxin Zhang, Suan Lee

**Affiliations:** 1Jewxon Intelligent Technology Co., Ltd., No. 293, Gongren East Road, Jiaojiang District, Taizhou 318000, China; 2School of Computer Science, Semyung University, Jecheon 27136, Republic of Korea

**Keywords:** binocular vision, robotic vision systems, bionic binocular-vision system, depth vision algorithms, machine vision, degrees of freedom artificial intelligence vision-based control

## Abstract

This paper describes a novel bionic eye binocular vision system designed to mimic the natural movements of the human eye. The system provides a broader field of view and enhances visual perception in complex environments. Compared with similar bionic binocular cameras, the JEWXON BC200 bionic binocular camera developed in this study is more miniature. It consumes only 2.8 W of power, which makes it ideal for mobile robots. Combining axis and camera rotation enables more seamless panoramic image synthesis and is therefore suitable for self-rotating bionic binocular cameras. In addition, combined with the YOLO-V8 model, the camera can accurately recognize objects such as clocks and keyboards. This research provides new ideas for the development of robotic vision systems.

## 1. Introduction

In the context of the new technological revolution and industrial transformation, the deep integration of artificial intelligence and robotics is essential for unleashing human productivity and accelerating the implementation of AI technologies. This profound empowerment of traditional industries effectively drives their transformation and upgrade [1,2,3]. Sensors perceive the external world as critical components of robots and provide essential information to support AI’s accurate decision-making [4,5]. Among these, vision sensors play a crucial role in the deep integration of AI and robotics. They are widely used, enabling robots equipped with vision systems to perceive and interpret their surroundings, thus allowing them to interact with the world and efficiently perform various tasks.

As one of the most crucial human organs, the eye plays a vital role in shaping our understanding of the world [6]. Given its significance, robots with vision systems can perceive and interpret their surroundings, enabling them to interact with the world and perform various tasks effectively [7,8,9]. The development of applications that combine machine vision as the eyes and artificial intelligence as the brain has been a focal point of research. In the 1990s, Yann LeCun pioneered the convolutional neural network (CNN) with his creation of LeNet-5, which endowed computers with the ability to learn and recognize image data [10]. Effectively, ImageNet, a large-scale image dataset, has acted as a catalyst for the development of computer vision, showcasing the impressive performance of deep learning in image recognition and laying the foundation for further research into vision-intelligent robots [11,12,13,14,15,16,17,18]. However, these applications typically utilize a single camera and focus on analyzing two-dimensional images on a flat plane. The material world humans inhabit is three-dimensional. The technology to capture and process three-dimensional information reflects a significant grasp of the target world and is a hallmark of intelligence [19]. Consequently, bionic eyes enable robots to obtain three-dimensional coordinates by mimicking the structure of the human eye, propelling artificial intelligence into a new phase of development. 

Monocular vision, binocular stereoscopic vision, and structured light technology have all been extensively researched; for more details, see Table 1. Monocular vision systems are known for their simple setup, fast response times, low power consumption, and cost efficiency. However, their main disadvantage is lower recognition accuracy. Representative companies in this field include Cognex, Honda, and Keyence. Instruments capable of three-dimensional measurement include binocular stereoscopic cameras [20], Time-of-Flight (TOF) cameras [21,22], structured light imaging cameras [23,24], holographic interferometry cameras [25], and more. Among these, the structure of binocular-vision cameras most closely resembles that of the human eye. Binocular stereoscopic vision technology uses two cameras to simulate the disparity principle of human eyes, obtaining planar and depth information from the real environment through triangulation. Companies like Luster (Bumblebee) and Fuayun (A100) represent the early stages of vision processing technology when algorithm and device precision were not very high; binocular-vision cameras do not emit light actively but rely entirely on two captured images to calculate depth, utilizing passive binocular imaging techniques. Consequently, structured light cameras were developed, using an infrared projection mode and binocular cameras to calculate the features projected onto object surfaces, thus significantly enhancing recognition precision compared to traditional stereoscopic cameras. However, this method requires high ambient light conditions and has a limited projection range, failing to function properly in reflective light conditions. Companies like Intel (Realsense), ORBBEC, and Microsoft (Kinect) represent this product category.

Binocular-vision cameras are more cost-effective compared to other 3D optical devices. They can also adapt to complex lighting conditions and capture dynamic scenes and moving targets. Consequently, binocular cameras are often used as bionic eyes and installed on mobile robots, where they have found widespread applications across various fields. In service robots, Qian implemented a binocular-vision system combined with the AdaBoost algorithm to detect facial regions in real time, facilitating tasks such as conversing with people, tracking faces, and coordinating mechanical arms to grasp objects [26]. In coal mine rescue robots, robots equipped with binocular-vision systems accurately collect information about collapses and obstacles within mines, providing critical feedback to rescue personnel to prevent secondary accidents [27]. Underwater rescue robots equipped with binocular stereovision, despite challenging and variable lighting conditions, demonstrate exceptional distance measurement capabilities and can perform underwater search and rescue tasks [28]. In an essential area of robotics—robotic arm vision—Sheng utilized the SURF algorithm to extract and match image feature points based on binocular vision, accomplishing the mechanical arm’s grasping of targets [29]. In the research on autonomous forestry robots for trunk distance measurement, Zhao developed a trunk-measuring system based on binocular vision theory using TI’s DaVincDM37x system. Zhao calculated three-dimensional information from images captured by the binocular cameras, obtaining accurate target location and distance measurements. To enhance the accuracy of obstacle detection algorithms in autonomous vehicles, Liang developed a binocular vision-based obstacle detection algorithm, which minimizes recognition errors and accurately measures the relative position between the vehicle and obstacles.

Binocular-vision technologies have been developed in recent years. Bionic binocular-vision technology mimics the visual mechanisms of humans and animals. However, most binocular cameras cannot rotate autonomously, thus missing autonomous flexibility. This paper primarily focuses on designing a binocular-vision system that can imitate human eye mechanisms, achieving flexibility similar to human eyes and automatically tracking targets for dynamic recognition, thereby providing a broader field of view, as shown in Figure 1. Eye movement is crucial for visual perception. Collecting more information, achieving a more comprehensive view, and eliminating blind spots all depend on the rotation of the eyes [30,31]. Eyes with degrees of freedom can smoothly track and locate targets, keeping the area of interest at the center of the frame and allowing quick responses to changes in the tracked target [32,33]. Particularly for robots performing target tracking tasks, when the target swiftly changes direction and the robot body cannot quickly turn, a binocular-vision system fixed on the mobile robot body is highly likely to lose track of the target. Therefore, researching a binocular-vision system with degrees of freedom is essential.

Basic binocular-vision systems have inherent limitations in wide-angle coverage and image quality, making it difficult for robots to meet performance requirements in complex and challenging environments. Therefore, this paper has developed two models of binocular-vision systems with degrees of freedom, including one model where only the camera rotates and another where both the axis and camera rotate together. The model where only the camera rotates is considered the ideal model. The proposed models can effectively improve the robotic vision performance, enhancing the system’s capabilities in challenging and complex environments, increasing its adaptability, and expanding its applications. Additionally, it is essential to note that traditional binocular-vision depth calculation methods, which rely on the principles of similar triangles, are no longer applicable when there are degrees of freedom in the binocular system. Thus, this paper also presents the depth calculation formulas corresponding to the two models.

## 2. Methods

### 2.1. Structural Design of Bionic Binocular Camera

To mimic the visual motion functions of human eyes, this study developed the JEWXON BC200 bionic binocular camera, as shown in Figure 2. This device consists mainly of precise miniature components, including the left eyeball and its associated motors. The motor for the up-and-down movement of the left eyeball is responsible for its vertical movement, simulating the up-and-down observation function of the human eye. In contrast, the left eyeball left/right movement motor handles horizontal movements, allowing the left eyeball to move from side to side, thus expanding the field of view. The right eyeball and its motors coordinate with the left eyeball to achieve binocular vision. The motor for the up and down movement of the right eyeball and the right eyeball left/right movement motor control the vertical and horizontal movements of the right eyeball, respectively. Both eyeballs are equipped with an IMU, which is responsible for collecting the attitude data of the eyeballs. The micro camera, embedded within the left and right bionic eyeballs, captures images and, along with the IMU data, transmits them in real time to the data acquisition sensors. These sensors are connected to the PCB via a cable, which transmits the data to the MCU for real-time processing. This lets the camera respond quickly in dynamic environments, capturing clear images and accurate eyeball attitude angles. After processing the data, the MCU communicates with the drive unit to perform the necessary operations. The drive is connected to the motors, controlling the multiple movement motors, allowing the left and right eyeballs to move up and down independently and left and right, significantly expanding the camera’s field of view. The PCB is designed for low power consumption, with a power usage of only 2.8 W, which is particularly important for mobile robots as low power consumption translates to longer operational times and higher efficiency. The PCB is mounted on the back of the bionic eyeballs and integrates data input/output interfaces. Through the coordinated work of these components, the JEWXON BC200 bionic binocular camera operates efficiently in dynamic environments, capturing high-quality images and providing precise eyeball attitude data, thus offering reliable visual perception capabilities for mobile robots.

### 2.2. Basis of Binocular Vision

Monocular cameras can capture two-dimensional images or videos but lack the ability to directly perceive depth information. As shown in Figure 3, on the same line, three different targets at varying distances project onto the same position in camera A. Therefore, camera A cannot distinguish which point is farther or closer based on the formed images. Due to the lack of a human eye-like stereoscopic sensing system and parallax for the target, monocular cameras are unable to directly acquire depth information. Thanks to studies of the human visual system, the researchers found that both eyes have different views of the same observation target. The brain could infer the depth of the target by comparing these two view differences.

The development of camera depth perception technology can be traced back to the 1980s. In 1998, based on the human eye’s mechanism, the Massachusetts Institute of Technology (MIT) started developing robot bionic eyes and created the Kismet robot [34]. The robot could interact by capturing facial expressions using two Charge-coupled Device (CCD) cameras. After that, an increasing number of researchers focused on developing robot vision [35,36,37], primarily centered around the research of bionic eyes composed of binocular cameras. Utilizing the parallax between the left and right cameras for the same observation target, binocular imaging techniques could calculate the distance from the camera to the target. As the target moves away from the camera, the parallax between the images seen by the left and right cameras decreases, and the difference increases as the target moves closer to the camera. To accurately calculate the distance to a target, it is necessary to calibrate the left and right cameras to determine their geometric relationship and distortion parameters. Subsequently, the feature points in the left and right images would be matched, and the depth information of the target would be obtained by utilizing the parallax of the image and the principle of triangulation.

A schematic diagram of the binocular-vision system is shown in Figure 4. In the ideal scenario of a robot bionic eye, two identical cameras are placed in parallel and simultaneously capture images. Due to the different positions of the two cameras (one left and one right), the image points formed by the same observation target in the left and right camera are not in the same position, resulting in a particular parallax. Based on the parallax between the left and right cameras, the depth information of the target distance from the binocular camera can be calculated by utilizing the principle of similar triangles.

p is the center of the observed target. $L_{1}$ and $L_{2}$ are the projection planes of the left and right cameras, respectively. $L_{1}\prime$ is the plane after rotating $L_{1}$, and $L_{2}\prime$ is the plane after rotating $L_{2}$. $o_{l}\prime$ and $o_{r}\prime$ are the optical centers of the two cameras, respectively. The installation height of the two cameras is A, the installation distance is B, and both cameras have a focal length of f. The horizontal length of the camera’s projection plane is C. To facilitate the understanding of the calculation in the presence of degrees of freedom, the two planes $L_{1}\prime$ and $L_{2}\prime$ are chosen as the reference planes. According to the triangle similarity theorem, it is easy to prove: $△px_{l}\prime x_{r}\prime ∼△po_{l}\prime o_{r}\prime$. The distance from fixed point p to $o_{l}\prime$ $o_{r}\prime$ is labeled Z′. Thus, there is the following relation:(1)$$\frac{B}{B-((B-C)+N_{2}+C-N_{1})}=\frac{Z^{\prime }}{f}$$The equation simplifies to:(2)$$Z^{\prime }=\frac{f*B}{N_{1}-N_{2}}$$From this, the target depth distance can be measured as:(3)$$Z=Z^{\prime }+f+A$$

f, A, and B can be obtained through measurement; thus, they are known parameters. By multiplying the pixel disparity of matched points with the real size of each pixel, the value of $N_{1}-N_{2}$ can be obtained. This is the depth calculation method employed for basic binocular vision.

### 2.3. Binocular Vision with Degrees of Freedom

However, in many specialized and complex environments, basic binocular-vision systems often fail to achieve the desired results; for example, those scenes that require wider viewing angles and higher image quality. Therefore, in this paper, two models of binocular-vision systems with degrees of freedom were built, including the model with only the camera rotating and the model of co-rotation of the axis and camera. Also, in the paper, the corresponding depth calculation formulas for these two models were provided.

#### 2.3.1. The Model with Only the Camera Rotating

Ideal binocular vision is shown in Figure 5. The binocular camera is free to rotate while the connecting axis is immobile perpendicular to the substrate.

A simple coordinate system was constructed in order to calculate the distance from the center p of the observed target to the substrate. α represents the angle of camera rotation, f denotes the camera focal length, A corresponds to the distance from the fixed point $o_{1}$ to the projection plane $L_{1}$, and C represents the horizontal length of the projection plane. $o_{l}$ was chosen as the origin coordinate (0,0). At the same time, clockwise rotation was considered positive, counterclockwise rotation was considered negative, and $|\alpha |<60^{\circ }$. The coordinates of the points of the left camera are shown in Table 2:

Let the line function of the line containing points $x_{l}$ and p be $g_{2}(x)=p_{1}x+q_{1}$. The slope $p_{1}$ of $g_{1}(x)$ could be calculated by the points $x_{l}\prime$ and $x_{l}$:(4)$$p_{1}=\frac{\cos\alpha *f-\sin\alpha *(N_{1}-\frac{C}{2})}{\sin\alpha *f+\cos\alpha *(N_{1}-\frac{C}{2})}$$By substituting point $o_{l}^{\prime }$ into $g_{1}(x)$, the bias term $q_{1}$ was obtained as:(5)$$\begin{array}{cc} q_{1} & =f(x)-p_{1}*x\\ & =Cos\alpha *f-\frac{Sin\alpha Cos\alpha *f^{2}-Sin^{2}\alpha *f*(N_{1}-C/2)}{Sin\alpha *f+Cos\alpha *(N_{1}-C/2)}\\ & =\frac{Cos^{2}\alpha *f*\left(N_{1}-C/2\right)+Sin^{2}\alpha *f*\left(N_{1}-C/2\right)}{Sin\alpha *f+Cos\alpha *\left(N_{1}-C/2\right)}\\ & =\frac{f*(N_{1}-C/2)}{Sin\alpha *f+Cos\alpha *(N_{1}-C/2)} \end{array}$$Similarly, let the points $x_{r}$ and *p* on the right lie on the line $g_{2}(x)=p_{2}x+q_{2}$. Similarly, the slope $p_{2}$ of the right camera was as follows:(6)$$p_{2}=\frac{Cos\alpha *f-sin\alpha *(N_{2}-C/2)}{Sin\alpha *f+Cos\alpha *(N_{2}-C/2)}$$The coordinates of $o_{r}$ could be calculated as (sinα∗f+B,cosα∗f). The bias term $q_{2}$ could be obtained by substituting point $o_{r}$ into the linear function $g_{2}(x)$:(7)$$q_{2}=\frac{f*(N_{2}-C/2)}{Sin\alpha *f+Cos\alpha *(N_{2}-C/2)}-B*p_{2}$$The coordinates of the intersection of $g_{1}$ and $g_{2}$ are the coordinates of point p.

#### 2.3.2. The Model of Co-Rotation of Axis and Camera

However, in practical application situations, the camera often needs to be fixed on top of a substrate with a certain thickness. Therefore, the model where the camera rotates alone and the axis does not rotate is the ideal case. The model of co-rotation of the axis and camera is more common and realizable. The schematic of the model of co-rotation of the axis and camera is shown in Figure 6.

A simple coordinate system was constructed in order to calculate the distance from the center p of the observed target to the substrate. α represents the angle of camera rotation, f denotes the camera focal length, A corresponds to the distance from the rotation point $o_{1}$ to the projection plane $L_{1}$, and C represents the horizontal length of the projection plane. $o_{1}$ was chosen as the origin coordinate (0,0), then $o_{2}$ coordinates were (B,0). At the same time, clockwise rotation was considered positive, counterclockwise rotation was considered negative, and $|\alpha |<60^{\circ }$. When the projection plane was parallel to the mounting substrate, α was considered $0^{\circ }$. The length of the projection plane and the mounting substrate were the same as when they are perpendicular, so the coordinates of the points on the left can be easily determined as shown in Table 3:

Let p and $o_{l}\prime$ be the function of this line, denoted as $f_{1}(x)=a_{1}x+b_{1}$. Since the points *p*, $o_{l}\prime$, $x_{l}$, and $x_{l}\prime$ were collinear, any two sets of coordinates could be substituted into the line equation to determine the function of this line. It followed that:(8)$$\begin{array}{cc} a_{1} & =\frac{Cos\alpha *A+Sin\alpha *(N_{1}-C/2)-\left(Cos\alpha *(A+2f)-Sin\alpha *\left(N_{1}-C/2\right)\right)}{Sin\alpha *A-Cos\alpha *\left(N_{1}-C/2\right)-\left(Sin\alpha *(A+2f)+Cos\alpha *\left(N_{1}-C/2\right)\right)}\\ & =\frac{-2f*Cos\alpha +2*Sin\alpha *(N_{1}-C/2)}{-2f*Sin\alpha -2*Cos\alpha *(N_{1}-C/2)}\\ & =\frac{Sin\alpha *(N_{1}-C/2)-f*Cos\alpha }{-Cos\alpha *(N_{1}-C/2)-f*Sin\alpha } \end{array}$$By substituting $a_{1}$ and $o_{L}^{\prime }$ into $f_{1}(x)$, $b_{1}$ could be determined as follows:(9)$$b_{1}=Cos\alpha *(A+f)-\frac{Sin\alpha *\left(N_{1}-C/2\right)-f*Cos\alpha }{-Cos\alpha *\left(N_{1}-C/2\right)-f*Sin\alpha }*Sin\alpha *(A+f)$$When $\alpha =0^{\circ }$, the expression for $f_{1}(x)$ could be simplified to:(10)$$f_{1}(x)=\frac{f}{\left(N_{1}-C/2\right)}x+A+f$$Similarly, the coordinates of each point of the right camera could be obtained, as shown in Table 4:

Compared to the left camera, the right camera differs mainly in the extra bias term B in the horizontal coordinate. Therefore, the calculation method for the slope $a_{2}$ in the corresponding function $f_{2}(x)=a_{2}x+b_{2}$ was similar, and it was as follows:(11)$$a_{2}=\frac{Sin\alpha *\left(N_{2}-C/2\right)-f*Cos\alpha }{-Cos\alpha *\left(N_{2}-C/2\right)-f*Sin\alpha }$$Since there was a bias term B, $b_{2}$ was:(12)$$\begin{array}{rr} b_{2} & =Cos\alpha *(A+f)-\frac{Sin\alpha *(N_{2}-C/2)-f*Cos\alpha }{-Cos\alpha *(N_{2}-C/2)-f*Sin\alpha }*(B+Sin\alpha *(A+f)) \end{array}$$Similarly, when $\alpha =0^{\circ }$, the function $f_{2}(x)$ was as follows:(13)$$f_{2}(x)=\frac{f}{-(N_{2}-C/2)}x+A+f-B*(\frac{f}{\left(N_{2}-C/2\right)})$$Point p was the intersection of functions $f_{1}(x)$ and $f_{2}(x)$, and the y-coordinate of point p represented the depth value of the observed target. The fixed point p had coordinates (x,y), $f_{1}(x)=a_{1}x+b_{1}$, $f_{2}(x)=a_{2}x+b_{2}$, and then the following relationship held:(14)$$\left\{\begin{array}{c} x=\frac{b_{1}-b_{2}}{a_{2}-a_{1}}\\ y=\frac{a_{2}b_{1}-a_{1}b_{2}}{a_{2}-a_{1}} \end{array}\right.$$When the camera performed up and down sweeps, there was an impact on depth perception. However, the vertical coordinates of the two cameras remained consistent, so the main impact was on the relative height, as shown in Figure 7.

Obviously, when there was a rotation angle along the vertical axis and the two cameras were parallel, the distance A relative to the mounting base plate ensured that the vertical rotation had little impact on the distance. Its effect was on the relative height of points $o_{1}$ and $o_{2}$. This relative height could be derived by constructing the function in the same way, and afterward, the distance between point p and the mounting base plate could be obtained.

## 3. Experiments

To assess the technological advantages of the bionic eye in next-generation vision devices, this project has procured mainstream industry depth cameras for experimental comparison. The cameras include Intel Realsense SR300 Depth Camera:Working distance: 0.3–2 m, Depth Field of View (FOV): H73°, V59°, D90°, Frame rate: VGA 30 fps, Power consumption: 1.8 W; ORBBEC Dabai DW Depth Camera:Working distance: 0.2–2.5 m, Power consumption: 4.0 W, Depth FOV: H79°, V55°, D88.5°, Operating temperature: 10 °C to 40 °C; Fuayun A100 Binocular Camera, Maximum resolution: 2560 × 960 30 fps, Angle FOV: H52°, V49°, Working distance: 0.2–8 m, Power consumption: 3.6 W; SXHDR 300 Binocular Camera, Maximum resolution: 1920 × 1080 30 fps, Angle FOV: H80°, V92°, Working distance: 0.2–5 m, Power consumption: 4.7 W; JEWXON BC200 Bionic Eye Binocular Camera, Maximum resolution: 3296 × 2512 30 fps, Static angle FOV: H65°, V51°, Best dynamic angle FOV: H206°, V192°, Baseline Length: 35 mm, Distortion: <1%, Built-in IMU, Working distance: 0.2–10 m, Low Power consumption: 1.8 W. The experimental results comparing the performance of these four cameras with the project-developed BC200 bionic eye binocular camera are illustrated in Figure 8.

During the preliminary operational tests, it was found that the Intel SR300 depth camera and the ORBBEC Dabai DW depth camera had similar specifications, leading to the decision to phase out the SR300 camera. Additionally, the SXHDR 300 binocular camera was found to be inferior to the Fuayun A100 binocular camera in terms of resolution, viewing angle, and power consumption, resulting in the elimination of the SXHDR 300 binocular camera from further consideration.

### 3.1. Experimental Platform

To achieve optimal performance of the cameras, this experiment involves connecting the cameras to different system platforms as detailed in Figure 9. The JEWXON BC200 bionic eye binocular camera is connected via USB 3.0 to Nvidia’s Jetson Orin NX 16 GB embedded vision development board. The operating system used is Ubuntu 20.04, with CUDA version 11.4.19. The AI performance of the setup reaches 100 TOPS.

The Fuayun A100 binocular camera is connected via USB 3.0 to the Allwinnertech H6 embedded development board, which features a quad-core Cortex A53 processor, Mali-T720 GPU, and 2 GB LPDDR3 memory. The operating system used is Android 7.0, as shown in Figure 10.

The ORBBEC Dabai DW depth camera is connected via USB 3.0 to an ASUS FX50 laptop, which features an i7 CPU 12,700 H at 2.30 GHz, 16 GB of memory, and an RTX 3070 GPU. The operating system used is Windows 11, as shown in Figure 11.

### 3.2. Binocular Ranging Accuracy Experiment

Accurately obtaining distances in the world coordinate system is a major feature of binocular stereovision. For this experiment, a high-precision DELIXI brand laser rangefinder, model D100, was selected, which has a maximum measuring distance of 120 M, as shown in Figure 12. This device was used to measure the pose of the cameras, thus allowing for a comparison of the actual measurement accuracy of the cameras.

In this study, the laser rangefinder and the cameras being tested were aligned with a reference target (a chessboard) to perform depth imaging measurements. The lenses of the three cameras and the emitter of the laser rangefinder were kept at a horizontal level. The distances measured were as follows: the Fuayun A100 binocular camera was 1.328 M from the chessboard; the ORBBEC Dabai DW depth camera was 1.339 M from the chessboard; and the JEWXON BC200 bionic eye binocular camera was 1.338 M from the chessboard, as shown in Figure 13.

### 3.3. Camera Parameters and Error Testing

Since the ORBBEC Dabai DW depth camera comes factory-calibrated with built-in direct ranging capabilities, as shown in Figure 14, this experiment only compares the two binocular cameras. Table 5 presents the basic parameters of the two experimental cameras obtained through the Matlab Stereo Camera Calibrator tool. BC200 may deliver images with less distortion, which is beneficial for applications that require high precision and image fidelity. From the camera error experiments shown in Figure 15 and Figure 16, it is evident that the average error for the Fuayun A100 binocular camera is 3.53 pixels, while the JEWXON BC200 bionic eye binocular camera has an average error of 0.08 pixels.

### 3.4. Distance Testing

Each of the three cameras acquired images on their respective platforms, and after shooting, calibration, and correction, the following data were obtained: ORBBEC Dabai DW Depth Camera: The actual distance from the camera to the target object was 1.339 M, with the camera’s measurement data showing 1.269 M; Fuayun A100 Binocular Camera: The actual distance from the camera to the target object was 1.328 M, with the camera’s measurement data showing 1.015 M; JEWXON BC200 Bionic Eye Binocular Camera: The actual distance from the camera to the target object was 1.338 M, with the camera’s measurement data showing 1.299 M. The experimental results indicate that the JEWXON BC200 bionic eye binocular camera’s measurement accuracy is closer to the world coordinate system distance compared to the ORBBEC Dabai DW camera. However, the Fuayun A100 binocular camera recorded measurement data of 1.015 M, which shows a significant deviation from the actual distance, as depicted in Figure 17.

### 3.5. Bionic Binocular Camera Dynamic Viewing Angle Test

Due to the JEWXON BC200 bionic eye binocular camera’s capability to rotate freely, similar to human eyeballs, it maintains a wider field of view compared to other fixed-depth and binocular cameras, even when the camera itself is in a fixed position. This feature allows for more comprehensive visual coverage, as depicted in Figure 18. 

In dynamic environments, objects do not always align perfectly with the camera’s field of view, making it crucial to measure depth data at various angles. However, binocular-vision systems inherently have installation errors that inevitably affect the calculation of depth values. Additionally, the inability to guarantee perfectly equal rotation angles for the left and right cameras is one source of error. As the bionic binocular camera’s rotation angle increases, the distance error caused by pixel differences at each point also increases. Minor pixel position errors can lead to significant distance measurement errors, causing errors to increase with the target distance. Using a basic model alone cannot achieve dynamic visual measurement acquisition.

The bionic binocular camera conducts real-time online calibration of images, which allows for the rapid acquisition of camera pose and image data. After obtaining left and right disparity images, the system corrects and processes them into images with depth information. However, during the rotation process of the binocular camera, using only the camera rotation model (Moda1) can cause image distortion during multi-angle motion shooting. This leads to incorrect pixel alignment, causing errors in the entire panoramic image composition.

On the other hand, the model where both the axis and camera rotate together (Moda2) results in images that are excessively smoothed during the JEWXON BC200 bionic eye binocular camera’s motion, reducing distortion and errors. The resulting panoramic images are more detailed, and the degree of image warping is significantly improved.

Experiments show that traditional binocular cameras experience increasing errors as the rotation angle increases. Furthermore, the improved model demonstrates smaller errors compared to the traditional model, proving the stability and effectiveness of the new model, as seen in Table 6. The comparison and results of these models and their impact on image quality and accuracy can be seen in Figure 19.

### 3.6. Target Detection Algorithm

In the field of machine vision, YOLO-V8 is a lightweight and highly efficient visual detector based on deep learning and computer vision, making it particularly suitable for embedded devices with strict power consumption requirements. YOLO-V8 outperforms other detectors such as YOLO-V5 and YOLO-V7 in both accuracy and speed and is well-suited for capturing targets in various complex scenarios. As an advanced and user-friendly object detection algorithm, it has attracted wide attention from researchers and developers [38]. This novel algorithm has been broadly applied in various fields, including autonomous driving, pedestrian tracking, robotics, and industrial inspection [39]. Therefore, this paper applies YOLO-V8 in conjunction with image data obtained from the JEWXON BC200 bionic eye binocular camera. As seen in Figure 20, after applying the algorithm that involves joint rotation of the axis and camera, the rotation process of the binocular camera is stable and smooth. The images are stable without significant jello effects or blurring. The accuracy of measurement and recognition is improved; it can accurately identify objects such as clocks, laptops, keyboards, mice, cups, pedestrians, and so on in the real world. This demonstrates that the bionic eye binocular camera not only provides a wider field of view and more stable images but also achieves high precision in autonomous target tracking, recognition, and ranging when combined with YOLO-V8.

### 3.7. Comparison of Existing Robotic Bionic Eye Devices

Binocular cameras are discussed in the article “Robot Bionic Vision Technologies: A Review” [40], as shown in Table 7. The bionic eye developed by Zou Wei et al. [41] consists of two CCD cameras and stepper motors, which essentially achieve the movement functions of the human eye. However, using large stepper motors results in an excessively large device with high power consumption, which is detrimental to mobility and portability. The bionic eye developed by Chen et al. [42] incorporates both long-focus and short-focus lenses in each eye, along with a three-degree-of-freedom neck mechanism integrated with an IMU. Although this design enhances perceptual capabilities, the overall size of the device and the high power consumption of the neck mechanism’s motor make it unsuitable for mobile applications. The JEWXON BC200, an improvement over previous generations of bionic eyes [43], features 4K HD micro-cameras and IMUs in a more compact form. It also utilizes the latest magnetic levitation conduction technology, allowing the eyeballs to rotate without cable interference, thus increasing the range of motion. Despite the significant reduction in overall power consumption, further improvements are needed to achieve additional functions such as head and neck movement.

### 3.8. Experimental Summary

The bionic eye can mimic the natural movements of the human eye and provide a wider field of view. When applied to humanoid robots, this enhanced vision technology significantly improves the robot’s visual perception capabilities, making it easier to track targets and make autonomous decisions in complex environments. The avoidance of repetitive head-turning motions allows for quicker responses in complex and dynamic scenarios. This paper introduces an innovative design for a bionic binocular-vision system, aimed at overcoming the limitations of existing technology by incorporating a model of binocular-vision systems with degrees of freedom, thereby enhancing the robot’s visual performance. Two models were developed: one involving only camera rotation and the other involving the rotation of both the axis and the camera. Various depth vision cameras were selected to conduct experiments on measurement accuracy and image quality. These experiments demonstrate that the designed bionic binocular-vision system and models not only simulate the flexible movement of human eyes but also significantly enhance the visual processing capabilities in dynamic environments by introducing degrees of freedom. The contributions of the proposed degree-of-freedom binocular-vision system models to the fields of machine vision and bionic eyes are as follows:Camera Specifications: JEWXON BC200 Bionic Eye Binocular Camera: Features a maximum resolution of 3296 × 2512 at 30 fps, Static angle FOV: H65°, V51°, Best dynamic angle FOV: H206°, V192°, integrated IMU, a working distance of 0.2–10 m, and a power consumption of 2.8 W. This camera surpasses the control group’s other cameras in resolution, maximum viewing angle, effective working distance, and energy efficiency.Vision System Combination: The JEWXON BC200, paired with the Jetson Orin NX, achieves an AI performance of 100 TOPS. This combination represents one of the more advanced embedded vision systems currently available, better suited for mobile robots due to its lower power consumption compared to PCs.Error Experiment: The JEWXON BC200 bionic eye binocular camera has the best distortion value, the average error is 0.08 pixels, and the performance is excellent, performing better than the Fuayun A100 standard binocular camera.Vision Ranging: The measurement accuracy of the JEWXON BC200 is closer to the world coordinate system distance compared to the ORBBEC Dabai DW depth camera. Using only the camera rotation model during the OpenCV panoramic image synthesis process can lead to image precision errors resulting in distortions and pixel misalignments that cause parts of the composite image to be missing. However, the combined rotation model of the axis and camera significantly improves image distortion, enhancing image precision and thus creating a more perfect composite image.Enhancing Robotic Vision: To make robotic vision more akin to human vision, the introduction of YOLO-V8 improves the autonomous recognition ability and recognition accuracy of the bionic eye. The BC200 can accurately identify objects such as clocks, laptops, keyboards, mice, cups, pedestrians, and so on in the real world. The bionic eye binocular camera not only provides a broader field of view and more stable images but also achieves autonomous target tracking and precise identification, surpassing the capabilities of fixed binocular and structured light cameras.

## 4. Conclusions

In this study, we developed the JEWXON BC200 bionic binocular camera and validated its effectiveness and innovation through comparative evaluation experiments with commonly used depth cameras and bionic binocular cameras. The JEWXON BC200 features a maximum resolution of 3296 × 2512 at 30 fps, a static angle FOV of H65° and V51°, and an optimal dynamic field of view of H206° and V192°. It integrates an IMU, operates within a distance of 0.2 m to 10 m, and consumes only 2.8 W, outperforming other cameras in terms of resolution, viewing angle, working distance, and energy efficiency. Combined with the Jetson Orin NX, which offers AI performance of up to 100 TOPS, the JEWXON BC200 forms one of the most advanced vision systems for embedded platforms, particularly suited for mobile robots. The camera demonstrated superior distortion performance with an average error of 0.08 pixels and achieved higher measurement accuracy compared to the ORBBEC Dabai DW depth camera. When using a model that rotates both the axis and the camera, the image distortion and precision errors observed in fixed installations were significantly reduced, resulting in more accurate and seamless composite images. Additionally, incorporating YOLO-V8 enhanced the camera’s ability to accurately identify objects such as clocks, laptops, and pedestrians, making the BC200 not only smaller and more energy-efficient but also more capable in autonomous target tracking and precise recognition, surpassing the capabilities of fixed binocular and structured light cameras.

## 5. Future Work Focus

A.The hardware platform for the binocular-vision system with degrees of freedom requires improvements, including adopting more advanced visual platforms, enhancing computational power while achieving lower operational power consumption, and increasing both installation and control precision. This will reduce the introduction of errors and enhance the accuracy of depth computation.B.While it is feasible to calculate distances using functional methods when the camera has degrees of freedom, it also introduces a considerable computational model that increases processing time; there is still room for future optimization.C.A solution is needed for the problem of absolute error variation with distance. One potential method to reduce errors is to attempt to replace the existing YOLO-V8 with a more precise and advanced target detection model.

The experiments conducted demonstrate that the designed bionic eye binocular-vision system offers better target depth calculation performance and a broader field of view compared to traditional binocular-vision systems. The experimental results confirm that the designed bionic binocular-vision system achieves flexible movements similar to human eyes while enhancing image quality and depth calculation accuracy during motion. This advancement not only enhances humanoid robots’ adaptability and application range in complex environments but also provides a crucial theoretical basis and technical pathway for the future development of robotic vision systems.

## Figures and Tables

**Figure 1 biomimetics-09-00371-f001:**
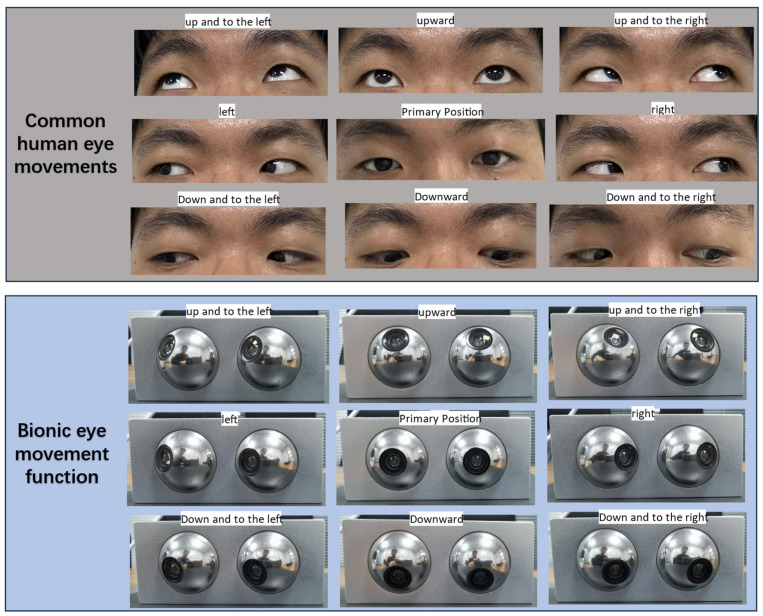
Bionic binocular-vision system designed based on the principles of simulating human eye movements.

**Figure 2 biomimetics-09-00371-f002:**
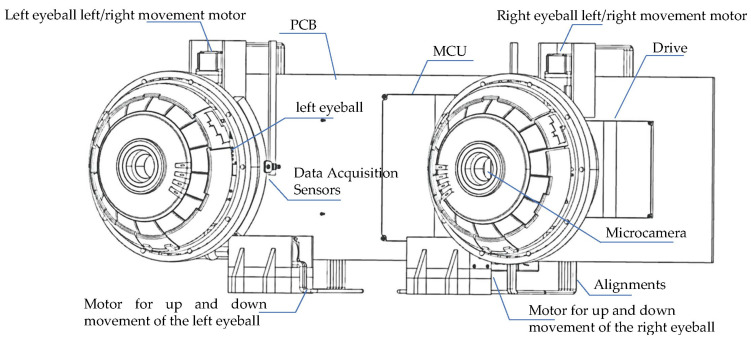
JEWXON BC200 Bionic eyes structure diagram.

**Figure 3 biomimetics-09-00371-f003:**
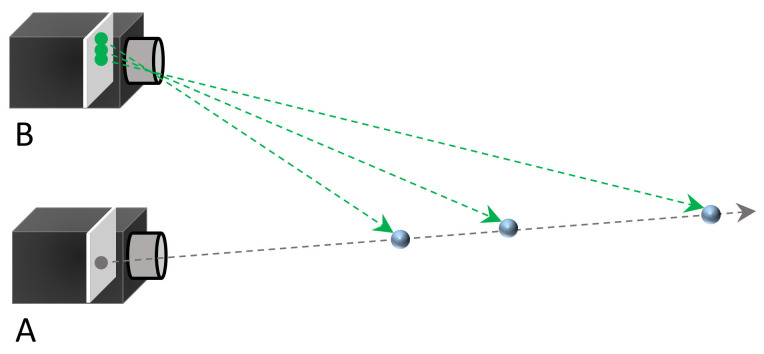
The schematic of the inability of the monocular camera to determine depth.

**Figure 4 biomimetics-09-00371-f004:**
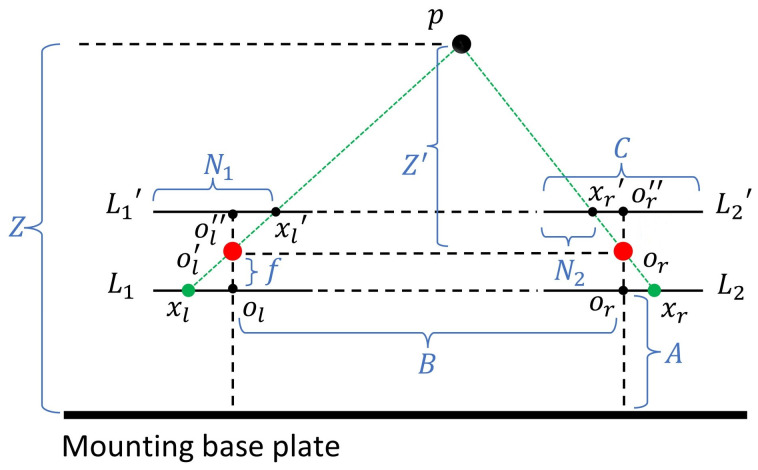
The schematic diagram of binocular-vision system.

**Figure 5 biomimetics-09-00371-f005:**
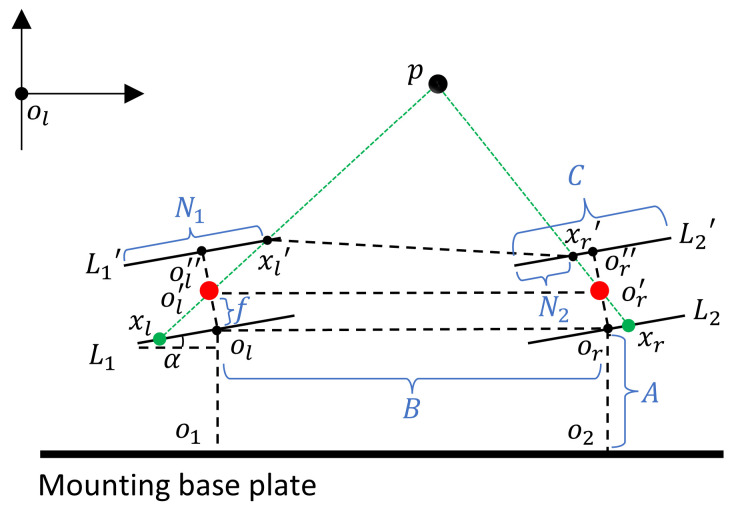
The schematic diagram of the model with only the camera rotating.

**Figure 6 biomimetics-09-00371-f006:**
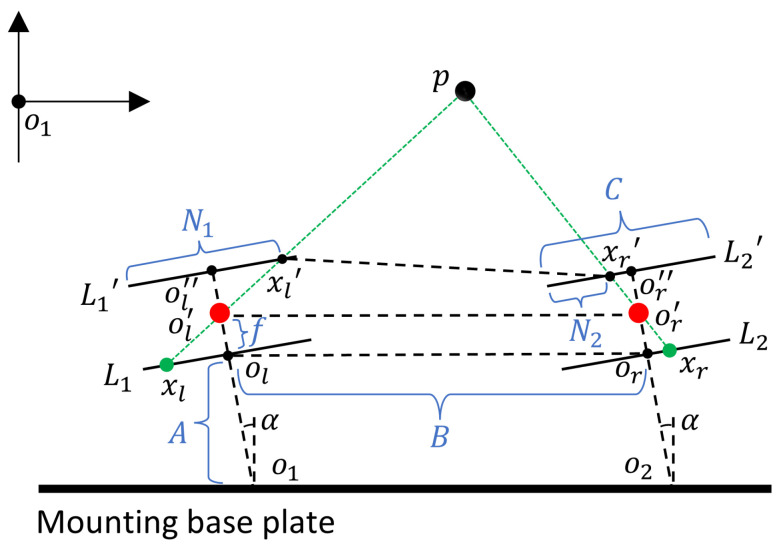
The schematic of the model of co-rotation of axis and camera.

**Figure 7 biomimetics-09-00371-f007:**
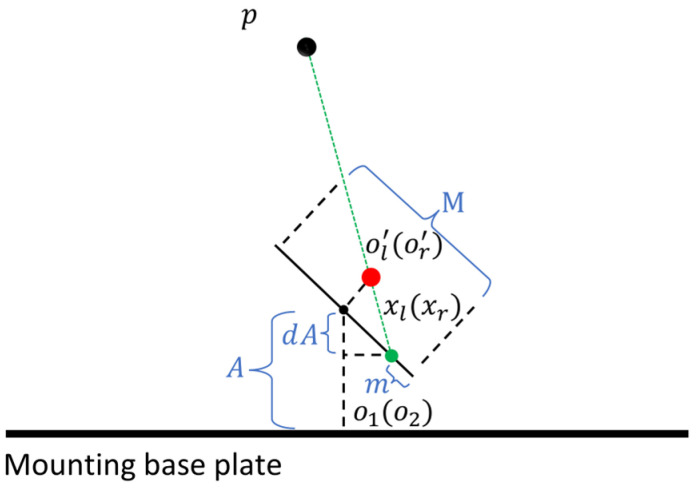
The relationship between the eyeball and the observation point *p* during vertical movement.

**Figure 8 biomimetics-09-00371-f008:**
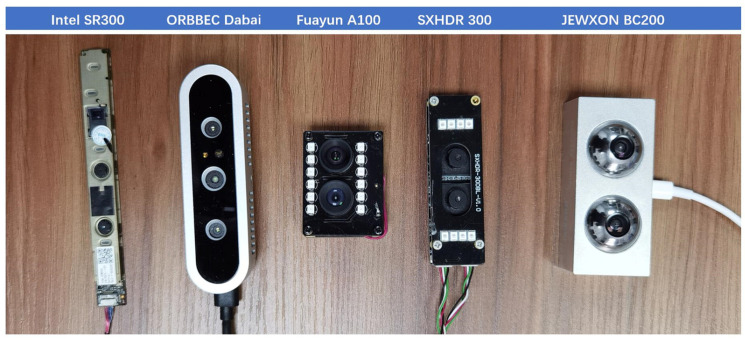
Camera evaluation photos.

**Figure 9 biomimetics-09-00371-f009:**
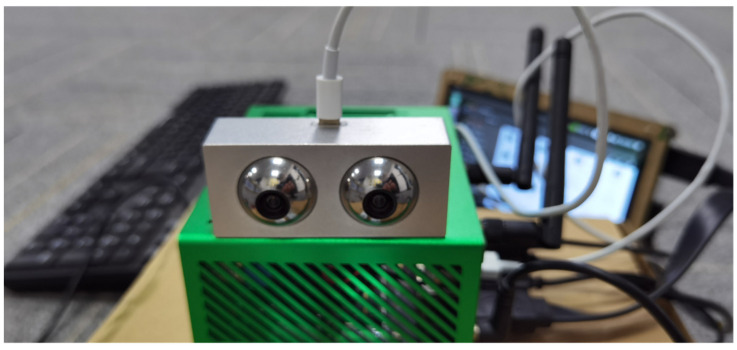
JEWXON BC200 bionic eye binocular camera system platform.

**Figure 10 biomimetics-09-00371-f010:**
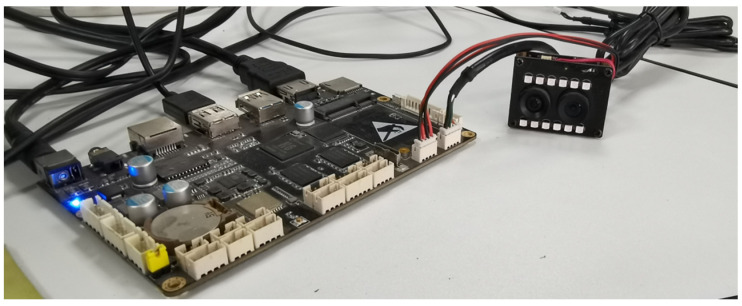
Fuayun A100 binocular camera system platform.

**Figure 11 biomimetics-09-00371-f011:**
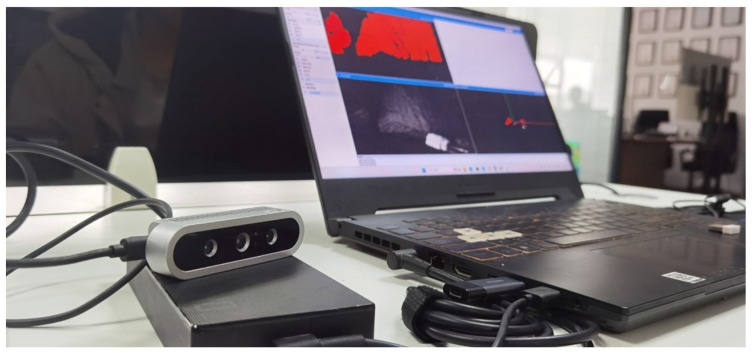
ORBBEC Dabai DW depth camera system platform.

**Figure 12 biomimetics-09-00371-f012:**
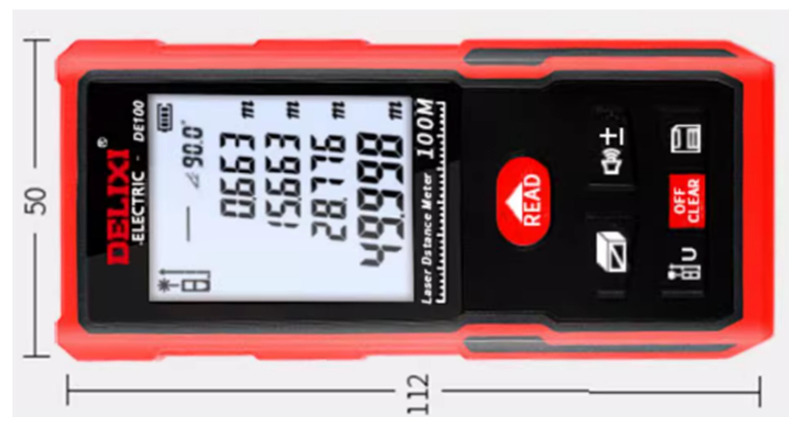
Laser rangefinder.

**Figure 13 biomimetics-09-00371-f013:**
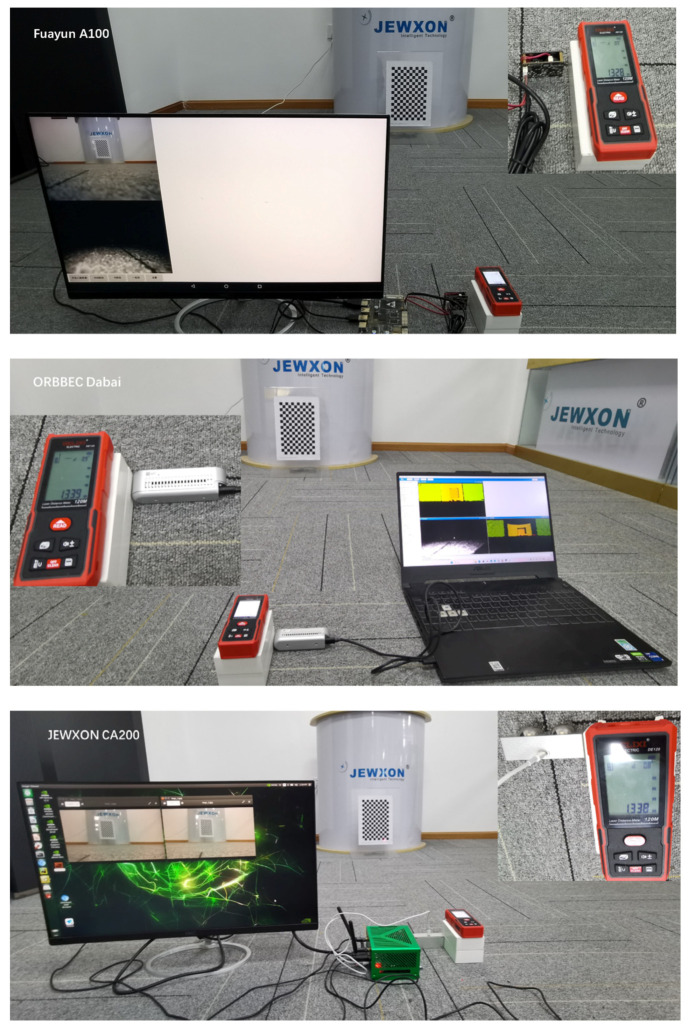
Reference to the actual distance from the target to the test camera, field experimental environment diagram.

**Figure 14 biomimetics-09-00371-f014:**
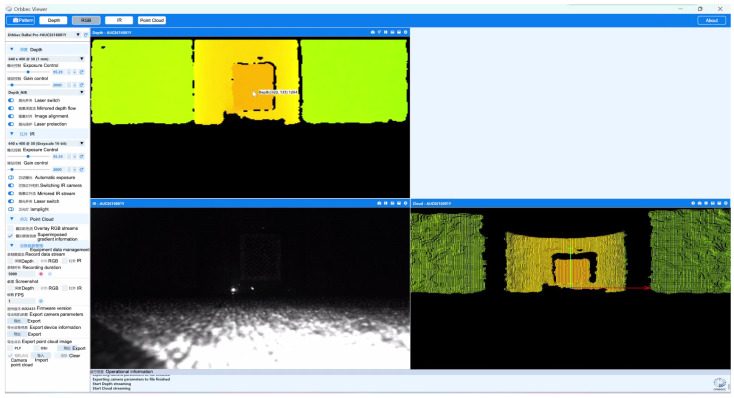
ORBBEC Dabai DW depth camera ranging experiment.

**Figure 15 biomimetics-09-00371-f015:**
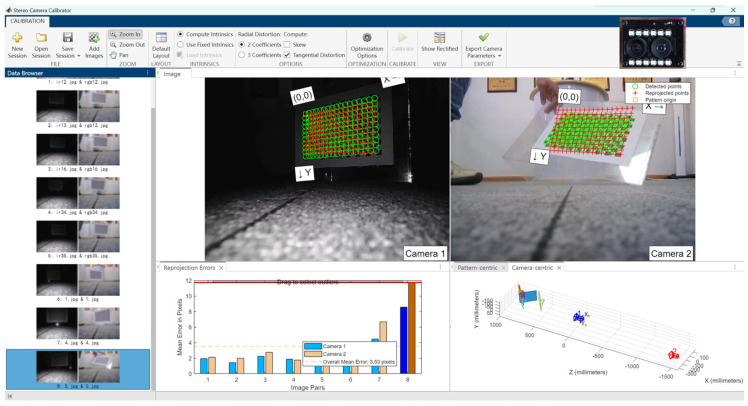
Fuayun A100 binocular camera error experiment. Reprojection Errors Image (Image Pairs 8): **left**: camera 1, **right**: camera 2.

**Figure 16 biomimetics-09-00371-f016:**
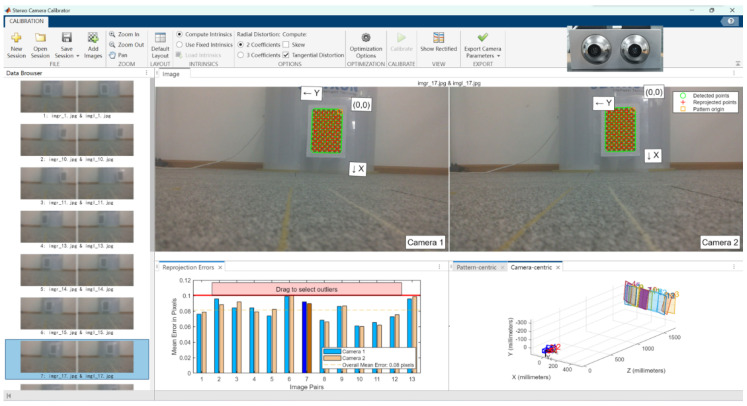
JEWXON BC200 bionic eye binocular camera error experiment. Reprojection Errors Image (Image Pairs 7): **left**: camera 1, **right**: camera 2.

**Figure 17 biomimetics-09-00371-f017:**
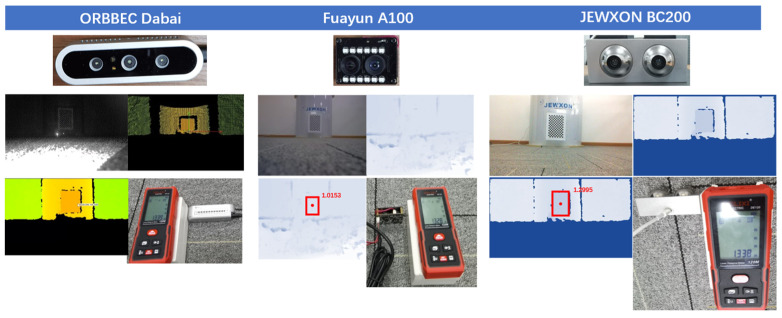
Display of measurement accuracy for each camera.

**Figure 18 biomimetics-09-00371-f018:**
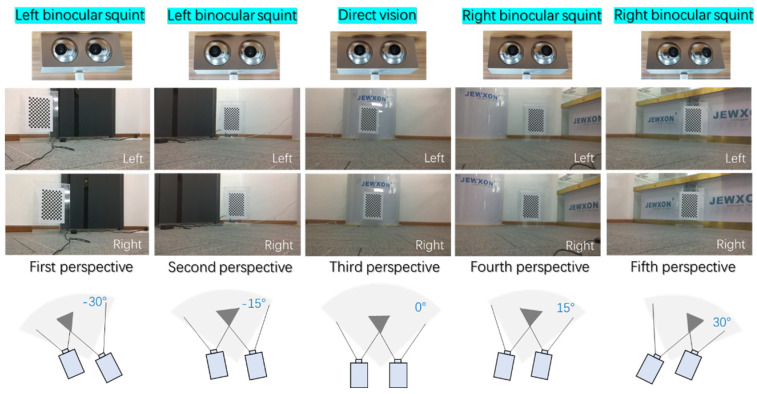
Multi-angle testing of the JEWXON BC200 bionic eye binocular camera after fixed installation. This test evaluates the performance of the JEWXON BC200 bionic eye binocular camera in a fixed installation, focusing on its ability to cover multiple angles due to its internal mechanisms that simulate human eye movements. The testing aims to demonstrate how the camera maintains a comprehensive field of vision across various orientations and conditions.

**Figure 19 biomimetics-09-00371-f019:**
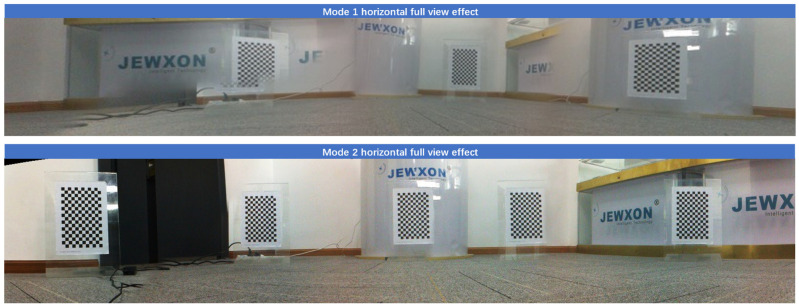
Panoramic synthesis comparison: only camera rotates model vs. axis and camera rotate model.

**Figure 20 biomimetics-09-00371-f020:**
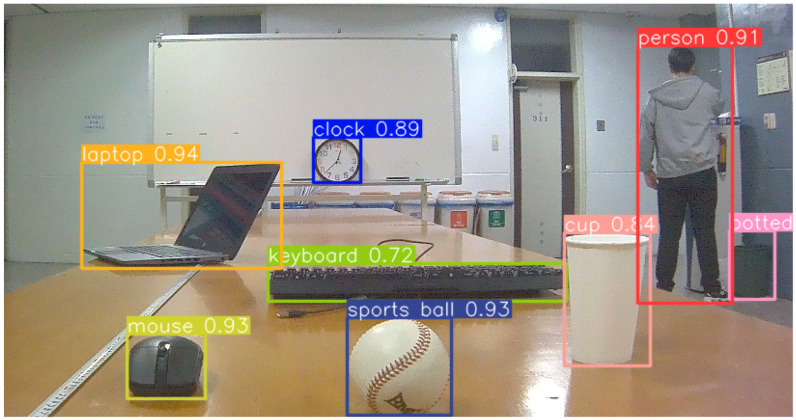
Bionic eye binocular camera combined with YOLO-V8 target detection and recognition effect.

**Table 1 biomimetics-09-00371-t001:** Performance comparison table of mainstream visual technology.

Technology Category	Monocular Vision	Binocular Stereo Vision	Structured Light	Bionic binocular Vision
Product pictures	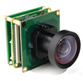	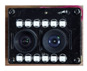	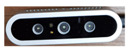	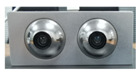
Technology principle	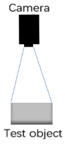	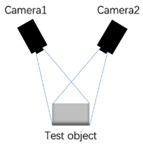	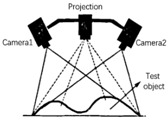	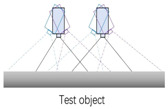
Principle of work	Single camera	Dual camera	Camera and infrared projection patterns	Autonomous motion dual camera
Response time	Fast	Medium	Slow	Algorithm determines speed
Weak light	Weak	Weak	Good	Weak
Bright light	Good	Good	Weak	Good
Identification precision	Low	Low	Medium	Algorithm determination accuracy
Resolving capability	High	High	Medium	High
Identification distance	Medium	Medium	Very short	Medium
Operation difficulty	Low	High	Medium	High
Cost	Low	Medium	High	High
Power consumption	Low	Low	Medium	Low
Disadvantages	Low recognition accuracy, poor dark light	Dark light features are not obvious	High requirements for ambient light, short recognition distance	Automatic target tracking, super wide viewing Angle
Representative company	Cognex, Honda, Keyence	LeapMoTion, Fuayun	Intel, Microsoft, ORBBEC	Huawei, Eyevolution, Jewxon

Data sources: Official websites of related products and patent databases.

**Table 2 biomimetics-09-00371-t002:** The coordinates of the points.

Point	x-Coordinate	y-Coordinate
$o_{l}$	0	0
$o_{l}\prime$	sinα∗f	cosα∗f
$o_{l}^{\prime \prime }$	2sinα∗f	2cosα∗f
$x_{l}$	$-\cos\alpha *\left(N_{1}-\frac{C}{2}\right)$	$\sin\alpha *\left(N_{1}-\frac{C}{2}\right)$
$x_{l}^{\prime }$	$2\sin\alpha *f+\cos\alpha *\left(N_{1}-\frac{C}{2}\right)$	$2\cos\alpha *f-\sin\alpha *\left(N_{1}-\frac{C}{2}\right)$

**Table 3 biomimetics-09-00371-t003:** The coordinates of the points of the left camera.

Point	x-Coordinate	y-Coordinate
$o_{1}$	0	0
$o_{l}$	sinα∗A	cosα∗A
$o_{l}\prime$	sinα∗(A+f)	cosα∗(A+f)
$o_{l}\prime \prime$	sinα∗(A+2f)	cosα∗(A+2f)
$x_{l}$	$\sin\alpha *A-\cos\alpha *(N_{1}-\frac{C}{2})$	$\cos\alpha *A+\sin\alpha *(N_{1}-\frac{C}{2})$
$x_{l}\prime$	$\sin\alpha *(A+2f)+\cos\alpha *(N_{1}-\frac{C}{2})$	$\cos\alpha *A+\sin\alpha *(N_{1}-\frac{C}{2})$

**Table 4 biomimetics-09-00371-t004:** The coordinates of the points of the right camera.

Point	x-Coordinate	y-Coordinate
$o_{2}$	B	0
$o_{r}$	B+sinα∗A	cosα∗A
$o_{r}\prime$	B+sinα∗(A+f)	cosα∗(A+f)
$o_{r}\prime \prime$	B+sinα∗(A+2f)	cosα∗(A+2f)
$x_{r}$	$B+\sin\alpha *A-\cos\alpha *(N_{1}-\frac{C}{2})$	$\cos\alpha *A+\sin\alpha *(N_{1}-\frac{C}{2})$
$x_{r}\prime$	$B+\sin\alpha *(A+2f)+\cos\alpha *(N_{1}-\frac{C}{2})$	$\cos\alpha *(A+2f)-\sin\alpha *(N_{1}-\frac{C}{2})$

**Table 5 biomimetics-09-00371-t005:** Test group stereo camera data table.

Metric/Camera	Fuayun A100	JEWXON BC200	Comparison
Focal Length (mm)	Left: 417.0386, Right: 456.7638	Left: 595.6271, Right: 594.2564	BC200 has a longer focal length, suitable for long-distance shooting and providing a wider field of view.
Distortion Coefficients	Left: [−0.057164, 0.095798, 0.011117, 0.000282, 0], Right: [−0.006848, −0.054429, 0.010137, 0.001460, 0]	Left: [0.0403, −0.095, −0.001167, 0.000428, 0], Right: [0.0299, −0.0322, −0.000880, −0.002340, 0]	BC200 has overall lower distortion coefficients, indicating higher image quality and less optical distortion.
Image Size (pixels)	640 × 480	640 × 480	Equivalent test environment
Rotation Matrix Stability	Lefit: [326.8532231, 252.4517895] Right: [323.8062822, 256.9767981]	Lefit: [321.5142126, 181.3599961] Right: [303.7775542 183.9382323]	BC200 may deliver images with less distortion, beneficial for applications that require high precision and image fidelity.
Translation Matrix	[16.9321,0.0315, 1.4005]	[−48.7250, −0.2644, 1.8614]	BC200 larger *Z*-axis offset is suitable for long-distance imaging.

**Table 6 biomimetics-09-00371-t006:** Comparison effect table of two models.

Feature/Metric	Only Camera Rotates Model	Axis and Camera Rotate Model
Stability of Image	Moderate	High
Image Distortion	Significant	Minimal
Coverage Area	Limited	Extensive
Resolution Consistency	Consistent	Consistent
Suitability for Dynamic Scenes	Moderate	Excellent

**Table 7 biomimetics-09-00371-t007:** Comparison of existing robotic bionic eye devices.

Zou Wei et al. R&D Team [44]	Chen et al. R&D Team [42]	JEWXON BC200
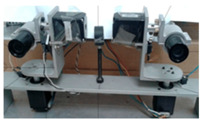	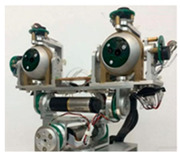	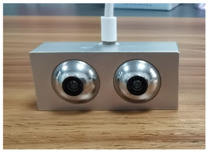
Features: The device consists of two CCD cameras and stepping motors, which can basically realize the movement function of the human eye.	Features: each eye contains two cameras (long-focus lens and short-focus lens) for simulating the perception of human eye features. And a 3-degree-of-freedom neck mechanism is designed with an integrated IMU.	Features: 4K HD mini-camera integrated in each eye, integrated IMU, and due to the latest levitation conduction technology used in the data collector, the eye rotates without being interfered by cables during the eye rotation, resulting in a larger angle of eye rotation.
Disadvantages: The model uses larger stepper motors, resulting in an excessively large product, and multiple large motors and loads running will inevitably lead to higher overall operating power consumption of the device, which is not conducive to mobility and portability.	Disadvantages: Although the motor controlling the rotation of the eyeball is reduced, the overall design of the bionic eye shape is too large and a larger motor is used for the neck mechanism, which leads to higher power consumption of the device and is not suitable for use with mobile devices.	Disadvantages: Although the overall use of mini cameras and mini motors, the overall power consumption drops a lot, but there is still room for improvement, such as the realization of the head and neck movement function.

## Data Availability

The data presented in this study are available upon request from the corresponding author.

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
