# Peer review of "Advancing the Robotic Vision Revolution: Development and Evaluation of a Bionic Binocular System for Enhanced Robotic Vision"

_biomimetics, 2024, doi:10.3390/biomimetics9060371_

Round 1
Reviewer 1 Report (New Reviewer)
Comments and Suggestions for Authors
This paper presents a “Advancing the Robotic Vision Revolution: 2 Development and Evaluation of a Bionic Binocular System for 3 Enhanced Robotic Vision”. The summary rating of the overall quality of the article is average. The following major suggestions:
1] The manuscript lacks a proper command of English and has a significant number of grammatical errors which should be significantly polished by a native English speaker.
2] In section I-Introduction is very concise, also references are very limited related to the work
3] The results should be more elaborated. It has been noticed that the discussion about the results are very precise in the paper. Could you provide more details on the selection process of depth vision cameras for conducting experiments on measurement accuracy and image quality
4] The abstract and conclusion are not written well in the paper. So rewrite the both with proper information. The abstract should not be more than 200-250 words.
5] How algorithms benefit from this level of performance, and how does it improve the overall functionality of the vision system
6] The extent of novelty demonstrated in the work is relatively limited. So add comparison table in the result section. How do these specifications of camera compare to existing technologies in the field
Comments on the Quality of English Language
Minor editing required
Author Response
- 1、Thank you very much for your suggestion, I have checked the full English and grammar and corrected the errors.
- 2、Already in the introduction section, references related to the bionic vision of this paper have been added, see: references 40-44.
- 3、Already in the experimental section, the process of camera selection has been added, as detailed in: paragraph 3.7
- 4、Thank you very much for your suggestions, the abstract and conclusions have been revised, see, Abstract and Conclusions section for details.
- 5、In the panoramic imaging experiments, we can clearly observe that the improved algorithm acquires the binocular symmetric image more accurately when the binocular camera rotates autonomously to capture the panoramic image, which effectively improves the image error caused by the binocular camera in the process of self-rotation.
- 6、A comparison of the effects of the bionic camera with existing visualization techniques has been added to the Experimental Summary section, see: Paragraph 3.7.

Reviewer 2 Report (New Reviewer)
Comments and Suggestions for Authors
In the introduction, other bionic binocular cameras are mentioned, but the paper does not clearly articulate the improvements of their vision system compared to those. If the main improvement is the autonomous rotating capability, the paper did not explain how this autonomous rotation is achieved. Additionally, the paper does not provide detailed information about the structure and components of the vision system, which would help to understand the design and functionality of the proposed vision system.
The experiments and the comparison of cameras in this paper are clear and well-organized. However, the comparison lacks an evaluation against other bionic binocular cameras.
Author Response
- Thank you very much for your suggestion. This paper is an upgraded and improved version of the previous generation of bionic eye module, which adopts a miniature camera and a micro-motor; the main innovations are the improvement of the existing binocular, and structured light and other vision technologies such as flexibility, wide angle of vision, and miniaturization, low power consumption.
- Thank you very much for your suggestion, a comparison with other bionic binocular cameras, has been added to the Experimental Summary section, see paragraphs 2.1 and 3.7 for details, as well as the Summary section.

This manuscript is a resubmission of an earlier submission. The following is a list of the peer review reports and author responses from that submission.
Round 1
Reviewer 1 Report
Comments and Suggestions for Authors
This manuscript presents a design of a bionic binocular vision system using YOLO-NAS. The manuscript misses some major points that are interesting for the reader. First, the authors should build a strong state of the art to highlight their contributions. Second, a deep discussion of the YOLO-NAS is missing, and how the authors could adapt it for their stated problem. Finally, the authors should discuss the limitations of the proposed method and report accuracy graphs with different vision situations. A flow chart should illustrate the generality of the proposed method.
Comments on the Quality of English LanguageNone.
Author Response
Response to Reviewer Comments
This manuscript presents a design of a bionic binocular vision system using YOLO-NAS. The manuscript misses some major points that are interesting for the reader. First, the authors should build a strong state of the art to highlight their contributions. Second, a deep discussion of the YOLO-NAS is missing, and how the authors could adapt it for their stated problem. Finally, the authors should discuss the limitations of the proposed method and report accuracy graphs with different vision situations. A flow chart should illustrate the generality of the proposed method.
- Thank you very much for your suggestions. The main intent of this paper is to design a bionic eye-inspired binocular camera and a model of a binocular automatic motion camera to improve image issues during movement. To address the YOLO issues you mentioned, we conducted multiple experiments and compared different versions of YOLO. We found that YOLO-V8 is more suitable for our experiments. Therefore, we have revised the paper accordingly. Please refer to the experimental section of the article for details.

Reviewer 2 Report
Comments and Suggestions for Authors
- In the evaluation section, it's important to conduct tests against benchmark models to showcase the improvements over other bionic binocular systems, particularly in demonstrating depth measurement precision.
- Given the reference to the phrase 'state-of-the-art...' in the title, it's essential to validate and verify our results against existing depth measurement precision standards.
Minor English language editing is required for a better presentation flow of this paper.
Author Response
Response to Reviewer Comments
- In the evaluation section, it's important to conduct tests against benchmark models to showcase the improvements over other bionic binocular systems, particularly in demonstrating depth measurement precision.
- Thank you very much for your patient feedback,This paper has been revised with new experiments that overcome the previously mentioned issues. For details, please see the Experiments section.
- Given the reference to the phrase 'state-of-the-art...' in the title, it's essential to validate and verify our results against existing depth measurement precision standards.
- Thank you very much for your suggestions. After conducting further experiments, this paper compares other depth cameras and includes additional depth measurement data. For details, please see the experimental section of the article.

Reviewer 3 Report
Comments and Suggestions for Authors
20240324 Review
Topic: Optimizing Robotic Vision: A State-of-the-Art Bionic Binocular 2 System for Depth Measurement Precision
In the paper, the authors presented two concepts for improving stereoscopic vision. They proposed the implementation of rotation of optical systems and mathematical models for calculating distances in space. The article should be supplemented in order to better understand the solution presented in it.
Note 1.
In the article, the authors write: Line168
For example, those scenes that require wider viewing angles and higher image quality.
Could you please explain what parameters are evaluated in the definition of "higher image quality"? The paper does not refer to any image quality parameters.
Note 2
In the article, the authors indicated an IMX219 sensor with a resolution of 1920 ∗ 1080 pixels. However, they did not specify the type of lens and its focal length. This makes it impossible to assess the field of view recorded on the sensors. This is an extremely important parameter from the point of view of assessing the imaging resolution and thus assessing the distance in the stereovision system. Please provide the imaging resolution at the distances indicated in the result tables, i.e. 300, 500, 1000 mm.
Note 3.
The authors proposed a rotation angle of α less than 60%. However, the ability to observe the p-point depends on the focal length of the lens and the angle of view of the lens. With such a large angle of rotation of the cameras, will the "p" point be seen on the camera on the left side of the stereo system? So for which lenses was this estimated? The question concerns Figures 3 and 4.
Note 4
Table 4 presents the results. The authors state that:
“The first column lists the measured distances with values of 30, 50, and 100 units. The second column displays the angle at which each measurement was taken, with 351 0°, 15°, and 30° listed. The third column, "Measured Distance," provides the actual measured distances, which vary slightly from the initial distances, indicating the results of the depth measurements. The fourth column lists the "Error" in the measurements compared to the true distances, and the fifth column shows the "Percentage Error" for each measurement.
How is the percentage error determined and between which values?
Distance = 100
Measure distance = 68.18, 73.20, 77.38 for three angles.
Percentage error = 0.82%, 1.20%, 1.62% ????
Note 5
How does the solution presented by the authors differ from the rotation of a standard integrated stereo imaging system by a selected angle and the acquisition of a series of 3D images during rotation?
Author Response
Response to Reviewer Comments
In the paper, the authors presented two concepts for improving stereoscopic vision. They proposed the implementation of rotation of optical systems and mathematical models for calculating distances in space. The article should be supplemented in order to better understand the solution presented in it.
Note 1.
In the article, the authors write: Line168
For example, those scenes that require wider viewing angles and higher image quality.
Could you please explain what parameters are evaluated in the definition of "higher image quality"? The paper does not refer to any image quality parameters.
- Thank you very much for your patient feedback,this paper primarily addresses the challenge of achieving a wider field of view in robot vision by simulating the movement characteristics of human eyes. Conventional binocular cameras are fixed, and attempting to mimic human eye squint—that is, rotating the two cameras independently—results in angular shifts and image distortion. This method significantly improves the quality of images captured by the camera in motion, as detailed in Section 3, "Experiments," Figure 1, and Table 5.
Note 2
In the article, the authors indicated an IMX219 sensor with a resolution of 1920 ∗ 1080 pixels. However, they did not specify the type of lens and its focal length. This makes it impossible to assess the field of view recorded on the sensors. This is an extremely important parameter from the point of view of assessing the imaging resolution and thus assessing the distance in the stereovision system. Please provide the imaging resolution at the distances indicated in the result tables, i.e. 300, 500, 1000 mm.
- This paper presents improvements to imaging devices, featuring a maximum resolution of 3296x2512 at 30fps. The static angle of field of view (FOV) is H65°, V51°, while the best dynamic angle FOV: H206°, V192°. It includes a built-in IMU, a working distance ranging from 0.2m to 10m, and a power consumption of 2.8W.
Note 3.
The authors proposed a rotation angle of α less than 60%. However, the ability to observe the p-point depends on the focal length of the lens and the angle of view of the lens. With such a large angle of rotation of the cameras, will the "p" point be seen on the camera on the left side of the stereo system? So for which lenses was this estimated? The question concerns Figures 3 and 4.
- The primary objective of this system is to provide a solution for situations where binocular cameras cannot rotate around a common pivot point simultaneously. The point "P" serves as a tracking point, and the main goal of the system is to observe while following this tracking point. In the diagrams, point "P" represents a historical position during movement.
Note 4
Table 4 presents the results. The authors state that:
“The first column lists the measured distances with values of 30, 50, and 100 units. The second column displays the angle at which each measurement was taken, with 351 0°, 15°, and 30° listed. The third column, "Measured Distance," provides the actual measured distances, which vary slightly from the initial distances, indicating the results of the depth measurements. The fourth column lists the "Error" in the measurements compared to the true distances, and the fifth column shows the "Percentage Error" for each measurement.
How is the percentage error determined and between which values?
Distance = 100
Measure distance = 68.18, 73.20, 77.38 for three angles.
Percentage error = 0.82%, 1.20%, 1.62% ????
- This paper has been revised with new experimental data, as detailed in Section 3, "Experiments".
Note 5
How does the solution presented by the authors differ from the rotation of a standard integrated stereo imaging system by a selected angle and the acquisition of a series of 3D images during rotation?
- Traditional stereo systems primarily encounter issues of image distortion and measurement accuracy when rotating the camera's position. The method described in this paper addresses these issues. For a comparison of the effects between the traditional approach and this new method, see Figure 18.

Reviewer 4 Report
Comments and Suggestions for Authors
1. In the introduction, references to old technologies are lengthy. For example, references to early CNNs or ImageNet do not provide special knowledge to those who read the paper. References to the latest technologies in stereo vision are required. The introduction requires comparisons with existing commercialized depth detection devices such as Kinect, RealSense, and Bumblebee stereo cameras.
2. It requires a description of the accuracy of the distance compared to the existing stereo vision method. If necessary, a comparison with the Bumblebee stereo camera, which was released more than a decade ago, is required.
3. Overall, it is a mathematical derivation of the geometrical relationship of the movement of the camera eyeball, but when it moves to the axis of the camera, it is difficult to find a difference from the method of finding depth from the existing stereo vision.
4. There seems to be no reason to use Yolo-nas as a method for detecting objects. Rather, it is necessary to improve the matching method between both images to obtain depth information for the entire front scene.
5. In the experiment, I recommend attaching an actual picture, not a graphic, of the bionic control system, which is the core of this paper. If you can take a picture of the target and the bionic control system used in the experiment together and see that the experiment has been conducted, you can confirm the practicality of the experiment.
6. In conclusion, to argue that the proposed system can provide a wider field of view, there should be more specific system configurations and experimental results.
Comments on the Quality of English LanguageThe sentences need to be refined a little more.
Author Response
Response to Reviewer Comments
- In the introduction, references to old technologies are lengthy. For example, references to early CNNs or ImageNet do not provide special knowledge to those who read the paper. References to the latest technologies in stereo vision are required. The introduction requires comparisons with existing commercialized depth detection devices such as Kinect, RealSense, and Bumblebee stereo cameras.
- Thank you very much for your suggestions. I have included the recommended introduction to stereo cameras in the introduction, as detailed in Table 1.
- It requires a description of the accuracy of the distance compared to the existing stereo vision method. If necessary, a comparison with the Bumblebee stereo camera, which was released more than a decade ago, is required.
- The Bumblebee stereo camera is simply a dual-camera stereo imaging system. Subsequently, structured light stereo camera systems, represented by Intel's RealSense, emerged. However, these systems fail to simulate the movement characteristics of the human eye. Therefore, this paper primarily focuses on addressing these issues.
- Overall, it is a mathematical derivation of the geometrical relationship of the movement of the camera eyeball, but when it moves to the axis of the camera, it is difficult to find a difference from the method of finding depth from the existing stereo vision.
- This paper does not completely deviate from existing stereo vision technologies but rather slightly modifies them to adapt to binocular cameras in motion. This adaptation allows robots to achieve better visual effects during movement. This technology is particularly suitable for autonomous binocular systems.
- There seems to be no reason to use Yolo-nas as a method for detecting objects. Rather, it is necessary to improve the matching method between both images to obtain depth information for the entire front scene.
- Thank you for your suggestions. I have revised the YOLO section as detailed in Figure 19.
- In the experiment, I recommend attaching an actual picture, not a graphic, of the bionic control system, which is the core of this paper. If you can take a picture of the target and the bionic control system used in the experiment together and see that the experiment has been conducted, you can confirm the practicality of the experiment.
- Thank you for your suggestions. I have added the images from the experiments, as detailed in Figures 1, 8, and 12.
- In conclusion, to argue that the proposed system can provide a wider field of view, there should be more specific system configurations and experimental results.
- Thank you for your suggestions. I have added the panoramic effect images, as detailed in Figure 18, along with the experimental conclusions.

Round 2
Reviewer 1 Report
Comments and Suggestions for Authors
The manuscript has been modified to some extent. For final acceptance, you can enhance the state of the art by mentioning some innovative AI applications in mdpi journals like:Residual neural networks for origin–destination trip matrix estimation from traffic sensor information and Notes on Bus User Assignment Problem Using Section Network Representation Method.
Comments on the Quality of English LanguageNone.
Author Response
- Thank you for your suggestion. I have thoroughly read the article "Residual Neural Networks for Origin–Destination Trip Matrix Estimation from Traffic Sensor Information." Given that this experiment primarily compares the performance of the innovative BC200 biomimetic binocular camera and the visual stability during binocular motion, your suggestion is very pertinent. In the future, I plan to integrate the BC200 into a robot's head and further innovate in conjunction with the methods proposed in "Residual Neural Networks for Origin–Destination Trip Matrix Estimation from Traffic Sensor Information."
Reviewer 3 Report
Comments and Suggestions for Authors
Thank you for answering my questions.
As additional information, the authors may provide the imaging resolution for different image recording methods. The resolution can also be specified on the imaging ranges defined by the authors. This will allow you to understand what objects are visible in the given range of 2-8 meters indicated by the authors.
You may find the following articles helpful:
1. 3D imaging methods in quality inspection systems. Proceedings of SPIE - The International Society for Optical Engineering, 2019, 11176, 111760L
Author Response
- Thank you very much for your suggestion. I have carefully read the article "3D Imaging Methods in Quality Inspection Systems" and thoroughly analyzed the experiments depicted in Figure 3. They bear a striking resemblance to the section illustrated by Figure 17 in my revised paper. Additionally, panoramic synthesis was achieved in Figure 18 instead of merely depth imaging. In the future, I intend to explore the application of BC200 in robot heads and conduct SLAM experiments in my next article. This will delve into the depth measurement effectiveness of robots in various scenarios during locomotion.